

# Coral reef origins of atmospheric dimethylsulfide at Heron Island, southern Great Barrier Reef, Australia

Hilton B. Swan[1,2,3], Graham B. Jones[1,2], Elisabeth S. M. Deschaseaux[1,2,3] and Bradley D. Eyre[1,3]

[1]Southern Cross University School of Environment, Science and Engineering, Lismore, New South Wales, Australia
[2]Marine Ecology Research Centre, Southern Cross University, Lismore, New South Wales, Australia
[3]Centre for Coastal Biogeochemistry, Southern Cross University, Lismore, New South Wales, Australia

*Correspondence to*: Hilton B. Swan (h.swan.11@scu.edu.au)

**Abstract**. Atmospheric dimethylsulfide ($DMS_a$), continually derived from the world's oceans, is a feed gas for the tropospheric production of new sulfate particles, leading to cloud condensation nuclei that influence the formation and properties of marine clouds, and ultimately the Earth's radiation budget. Previous studies on the Great Barrier Reef (GBR), Australia, have indicated coral reefs are significant sessile sources of $DMS_a$ capable of enhancing the tropospheric $DMS_a$ burden mainly derived from phytoplankton in the surface ocean; however, specific evidence of coral reef DMS emissions and their characteristics is lacking. By using on-site automated continuous analysis of $DMS_a$ and meteorological parameters at Heron Island in the southern GBR, we show that the coral reef was the source of occasional spikes of $DMS_a$ identified above the oceanic $DMS_a$ background signal. In most instances, these $DMS_a$ spikes were detected at low tide under low wind speeds, indicating they originated from the lagoonal platform reef surrounding the island, although evidence of longer range transport of $DMS_a$ from a 70 km stretch of coral reefs in the southern GBR was also observed. The most intense $DMS_a$ spike occurred in the winter dry season at low tide when convective precipitation fell onto the aerially exposed platform reef. This co-occurrence of events appeared to biologically shock the coral resulting in a seasonally aberrant extreme $DMS_a$ spike concentration of 45.9 nmol m$^{-3}$ (1122 ppt). Seasonal DMS emission fluxes for the 2012 wet season and 2013 dry season campaigns at Heron Island were 5.0 and 1.4 µmol m$^{-2}$ d$^{-1}$, respectively, of which the coral reef was estimated to contribute 4 % during the wet season and 14 % during the dry season to the dominant oceanic flux.

## 1 Introduction

Dimethylsulfide (DMS) is the major volatile sulfur compound released from the global oceans (Andreae and Raemdonck, 1983). The primary source of DMS is dimethylsulfoniopropionate (DMSP) which is a metabolite of many marine phytoplankton (Stefels, 2000). DMSP is also present in coral tissue, its symbiotic microalgae, and coral mucus (Broadbent and Jones, 2004). When DMS diffuses from the coral biomass to the water column it is then available for exchange across





the air-sea interface which is mostly driven by wind (Ho and Wanninkhof, 2016). The shallow water column over a coral reef has a lower thermal capacity than the open ocean so it is subject to enhanced heating by incident solar radiation (McGowan et al., 2010). This will lower the diffusivity resistance for mass transfer of DMS through the seawater surface film, as described by the Schmidt number, which is temperature dependent (Saltzman et al., 1993). This thermal effect,

which can enhance the air-sea exchange of DMS (Yang et al., 2011), is expected to be pronounced during daytime low tides over coral reefs when elevated atmospheric DMS ($DMS_a$) concentrations have been observed (Jones and Trevena, 2005). Additionally, DMS may be directly exchanged to the atmosphere from the coral surface if aerially exposed at low tide. These particular characteristics of coral reefs suggest that they could be 'hotspots' for production of $DMS_a$ oxidation products contributing to the sulfate component of new aerosol particles measured from the Great Barrier Reef (GBR) (Modini et al.,

2009; Vaattovaara et al., 2013). These new sulfate particles typically have hygroscopic properties that allow them to grow to a critical threshold diameter capable of acting as cloud condensation nuclei (CCN, $D_p$ ~100 nm), which can produce high-albedo low-level marine clouds with numerous small droplets that efficiently scatter sunlight and reflect it back to space (Ayers and Gillett, 2000) (Fig. 1a). Since these high-albedo clouds are capable of altering incident solar radiation, it was hypothesized almost thirty years ago that biological DMS production and its subsequent air-sea exchange leading to sulfate

aerosol and CCN might generate a climate feedback loop (Charlson et al., 1987); however, two decades of intensive Earth Systems research has provided a lack of evidence for DMS dominated control of CCN leading to global climate regulation (Quinn and Bates, 2011). Nevertheless, since DMS is an important trace gas that has the potential to play a significant role in the Earth's radiation budget, there is continuing interest in measuring DMS in the biosphere for the estimation of sea-to-air emission fluxes to assist climate and Earth System modeling (Lana et al., 2011).

Even though the total area of the world's coral reefs amount to ~0.2 % of the entire marine environment, the GBR is classed as the single largest living organism on Earth that stretches ~2,300 km along the NE coast of Australia, and covers an area of ~344,400 $km^2$ that can be seen from space (Hutchings et al., 2008). Considering these dimensions, the GBR is clearly a significant marine ecosystem that has been under-studied in comparison to the surface ocean for DMS production. Chamber experiments have provided evidence that *Acropora* spp. of branching coral can be a source of $DMS_a$ in the natural

reef environment. This genus of reef-building coral, which is dominant throughout the GBR (Wild et al., 2004), has been found to liberate DMS to the headspace of sealed chambers when submersed in seawater and bubbled with air (Fischer and Jones, 2012). Another chamber study that examined *Acropora* coral in filtered seawater detected DMS in the chamber headspace only when the coral was visibly coated in mucus or releasing mucus strands (Swan et al., 2016) which indicates that coral reefs are likely to be intermittent sources of $DMS_a$. There is currently insufficient data available from previous

GBR $DMS_a$ field studies to environmentally assess that recent chamber observation. The few previous field studies that have measured $DMS_a$ over the GBR (Broadbent and Jones, 2006; Jones and Trevena, 2005) employed gold-wool chemi-adsorption (Barnard et al., 1982; Kittler et al., 1992), a low-frequency grab-sampling technique. While the investigators of those previous field studies reported that the GBR appears to be a significant source of $DMS_a$, their low frequency sampling technique did not provide sufficient data to characterize coral reef DMS emissions. In this study we used an automated gas



chromatograph (GC) with a higher sampling frequency to examine short-term variations in $DMS_a$ over 2-3 week periods at a coral cay in the southern GBR. The objective of this study was to gather specific environmental evidence of coral-derived DMS emissions from the GBR, and to determine the intensity, frequency, and significance of those reef emissions in the wet and dry seasons at the Tropic of Capricorn.

## 5 2 Methods

### 2.1 Location and sampling procedures

The automated GC was used to continuously measure $DMS_a$ at the Heron Island Research Station (23.44°S, 151.91°E), situated ~80 km off the east Australian coastline, in the Capricorn-Bunker Group of southern GBR reefs (Fig. 2). Heron Island lies at the western end of a surrounding 27 $km^2$ lagoonal platform reef. The reef lagoon has a cover of ~15 % coral and

10 85 % permeable carbonate sands (Eyre et al., 2013). Two campaigns were conducted: 6-20 March 2012, and 18 July to 5 August 2013, providing a comparison of $DMS_a$ in the austral warmer wet (November to March) and cooler dry (April to October) seasons that are recognised at the Tropic of Capricorn. $DMS_a$ was sampled through Teflon™ tubing fixed to the roof of the Research Station laboratory; the intake was shielded from rain and had a clear line of sight to the reef flat ~100 m away. The automated GC had a cycle time of 26 min and was fitted with a pulsed flame photometric detector (Cheskis et al.,

1993). A custom-built autosampler controlled the GC, two gas valves and a cryogenic trap. This GC system had a sample collection time of 14.4 min and a cycle time of 26.0 min. Specified $DMS_a$ concentrations are reported at the mid-time of the collection period. The limit of reporting was 0.1 nmol $m^{-3}$ and the relative measurement uncertainty at the 95 % confidence level was 13 %. Details of the configuration, operation, calibration and measurement uncertainty of the automated GC $DMS_a$ sampling system are reported elsewhere (Swan et al., 2015).

A wireless automatic weather station (AWS, model XC0348, Electus Distribution, Rydalmere NSW, Australia) was mounted above the roof-line of the research station laboratory within 1 m of the air intake used to sample $DMS_a$. This AWS provided data for wind speed (WS), wind direction (WD), rainfall, indoor and outdoor air temperature, humidity and barometric pressure, which were logged at 15 minute intervals. The accuracy of the AWS was specified as: WS ± 1 m $s^{-1}$ (WS < 10 m $s^{-1}$) and ± 10 % (WS > 10 m $s^{-1}$), relative humidity ± 5 %, temperature ± 1°C and pressure ± 3hPa. Water vapour

mixing ratios were calculated from the AWS data using reported formulas (Vaisala, 2013). A pyranometer was placed on the laboratory roof next to the AWS (HOBO pendant sensor, Onset Computer Corp., Bourne, MA, USA). It logged solar irradiance (spectral range 300 - 1100 nm) every 15 min and was specified by the manufacturer to have an upper light intensity limit of 323,000 lumens $m^{-2}$ (Lux), which is equivalent to ~6000 µmol $m^{-2}$ $s^{-1}$ or 3.613 x $10^{21}$ photons $m^{-2}$ $s^{-1}$. The photon flux (µmol $m^{-2}$ $s^{-1}$) measured with this sensor was converted to radiometric energy (J $m^{-2}$ $s^{-1}$ or W $m^{-2}$) using a 0.342

J µmol$^{-1}$ multiplication factor (Thimijan and Royal, 1982). To complement the AWS data, daily backward trajectory air parcel information was obtained using the NOAA Air Resources Laboratory HYSPLIT transport and dispersion model (Stein et al., 2015). Mean surface layer pressure synoptic charts at 00:00 UTC and 12:00 UTC were also obtained from the



Australian Bureau of Meteorology (BoM). Tidal information was sourced from predictions provided by the National Tidal Unit of the Australian Bureau of Meteorology. Several site-specific observations of seawater drainage from the Heron Island reef flat showed that low tides consistently occurred +1.25 h after the predicted times; tide times were adjusted accordingly. All specified dates and times are Australian Eastern Standard Time.

5    ## 2.2 Flux calculations

Seasonal DMS emission fluxes for the wet season (14 day) and dry season (18 day) campaigns at Heron Island were calculated using the photochemical ambient mass balance equation applied by *Ayers et al.*, (1995) under clean marine conditions:

$$\frac{d[DMS]}{dt} = \frac{F_{DMS}}{H} - K[OH][DMS] + \frac{E_v([DMS_t]-[DMS])}{H}$$

(1)

where $F_{DMS}$ is the flux of DMS; $[DMS]$ is the mean concentration of $DMS_a$ in the marine boundary layer (MBL); $[DMS_t]$ is
the concentration of $DMS_a$ in the atmospheric transition layer or entrainment zone, $E_v$ is the MBL - lower troposphere entrainment velocity; $H$ is the mean depth of the MBL; $[OH]$ is the diurnally averaged concentration of hydroxyl radical; and $K$ is the first order overall rate constant for reaction of OH with DMS. The HYSPLIT transport and dispersion model was used to obtain midday mixed layer depths, which were averaged for each season to provide input values for $H$. Seasonal contributions of the coral reef to $F_{DMS}$ at Heron Island were estimated from the difference between fluxes calculated using
mean and median $DMS_a$ dataset concentrations.

## 3 Results and Discussion

### 3.1 Wet season 2012

Spikes of $DMS_a$ were occasionally observed that could be attributed to Heron Island reef flat DMS emissions. These spikes were usually detected at low tide under low WS, indicating that the prevailing SE trade winds (Fig. 2) are likely to have
diluted some $DMS_a$ spikes from the coral reef to the extent that they were indistinguishable from the ocean-derived $DMS_a$ background signal. During the wet season campaign the oceanic background $DMS_a$ signal was clearly coupled to WS (Huebert et al., 2010), while four distinct spikes of $DMS_a$ (10, 14, 16-17 March) were notably uncoupled from WS (Fig. 3a). The first of these $DMS_a$ spikes on 10 March occurred between 3:33 to 6:08 under low WS of 0 - 0.7 m s$^{-1}$, where the circumstances surrounding this event implicates the Heron Island reef flat as the source of the $DMS_a$ spike. In the early
morning of the 10 March back trajectories show that high altitude (+1000 m) continental air was directed to Heron Island





(Fig. 4). This dry free tropospheric non-marine air flow was characterised by a rapid drop in the water vapour mixing ratio from 17.4 to 13.0 g kg$^{-1}$, and also resulted in the lowest $DMS_a$ wet season concentration of 0.6 nmol m$^{-3}$ which occurred shortly after high tide under calm conditions. As the tide dropped, the $DMS_a$ steadily increased, and in the period 10 March 3:59 to 5:42 the anemometer was becalmed. Between 3:07 and 3:33 the $DMS_a$ rapidly increased from 2.1 to 6.2 nmol m$^{-3}$ and, thereafter, five consecutive $DMS_a$ concentrations of ~6 nmol m$^{-3}$ were recorded over a period of nearly two hours when WS was 0 m s$^{-1}$. The low tide at 4:50 occurred near the middle of this period of no air movement. A peak $DMS_a$ concentration of 6.4 nmol m$^{-3}$ was recorded at 5:16; however, it decreased to 2.6 nmol m$^{-3}$ by 7:00 with the onset of a southerly change at 6:36 when WS sharply increased from 0.3 to 2.4 m s$^{-1}$. The WS continued to increase in strength, and by 10 March 7:26 $DMS_a$ had returned to the background concentration of 2.2 nmol m$^{-3}$ (Fig. 3b). These observations indicate that the relatively low WS of 2.4 m s$^{-1}$ was sufficient to rapidly dilute the DMS plume with marine air containing less $DMS_a$. The meteorological conditions under which this $DMS_a$ spike occurred provide compelling evidence that it was a point source emission derived from the lagoonal platform reef surrounding Heron Island.

Rain enhanced air-sea exchange of DMS from the Heron Island reef flat was observed on 14 March 2012 at 21:12, which resulted in a peak $DMS_a$ concentration of 10.6 nmol m$^{-3}$. The increase in $DMS_a$ began in the evening at 19:02 and returned to the background level at midnight. At the start of this event the WS was between 3 and 4 m s$^{-1}$, and the background oceanic-derived $DMS_a$ was ~4 nmol m$^{-3}$. There had been no rain during the day; however, between 19:28 and 22:04 a local convection generated storm delivered a total of 12 mm of precipitation which fell during a 0.8 m low tide. The $DMS_a$ rose sharply from 3.9 to 10.6 nmol m$^{-3}$ in a 2.5 h period as a result of these two simultaneous events. Three steps were observed in this $DMS_a$ spike and appeared to be linked to variations in rainfall intensity and WS. After the onset of the rainfall there was a pause in the rain and the WS increased to 5.1 m s$^{-1}$ resulting in a pause in the rise of $DMS_a$; the WS then dropped to 2 m s$^{-1}$ as the precipitation intensity increased, resulting in the next rise in the $DMS_a$ spike. When the rainfall was most intense the WS had dropped to 1.4 m s$^{-1}$ and this resulted in the peak $DMS_a$ concentration which also occurred at the tidal minimum. As the rain eased, the WS again increased and the $DMS_a$ concentration rapidly decreased with the rising tide, returning to the background concentration of 4.3 nmol m$^{-3}$ at 1:05 on 15 March. This combination of a 0.8 m low tide coupled with a 2.5 h convective rainfall event resulted in an increase of 6 nmol m$^{-3}$ above the prevailing background ocean $DMS_a$ signal. The peak $DMS_a$ concentration which co-occurred with the lowest recorded WS was apparently due to reduced mixing of the rain-enhanced local $DMS_a$ plume from the reef with background marine air containing less $DMS_a$. Later, on 15 March between 12:40 and 14:50, another local convection storm delivered 7.2 mm of rain under a higher average WS of ~4 m s$^{-1}$. This later rain event did not coincide with low tide and did not induce a detectable $DMS_a$ spike from the Heron Island reef flat (Fig. 3c). The third $DMS_a$ spike which occurred between 21:20 16 March and 00:48 17 March (3.5 h duration), was observed as the WS abated from 3.4 to 0.7 m s$^{-1}$ indicating that it did not result from increased air-sea exchange. The background $DMS_a$ signal before and after this spike was, however, clearly decreasing with the abating WS (Fig. 3c). The peak $DMS_a$ concentration of 8.2 nmol m$^{-3}$ occurred at midnight on a 0.8 m low tide under a low WS of 1.7 m s$^{-1}$. These



meteorological conditions again implicate the reef flat surrounding Heron Island as the source of this $DMS_a$ spike, which was linked to the tidal minimum.

The highest recorded $DMS_a$ concentration of 11.5 nmol m$^{-3}$ during the wet season was detected on 16 March at 6:27. This $DMS_a$ spike lasted for 8.4 h, was a factor of 3.3 above the oceanic background level at its peak, and according to back

trajectories, occurred during a low-elevation marine air stream from a SE direction (Fig. 5). Unlike other observations, this $DMS_a$ spike started on a 2.5 m high tide and was rapidly decreasing by the time of the following 0.9 m low tide. WS observations indicate that the $DMS_a$ spike was not derived from wind enhanced air-sea exchange because $DMS_a$ increased from 3.5 to 11.5 nmol m$^{-3}$ as the WS eased from 4.1 to 3.1 m s$^{-1}$, and the $DMS_a$ peak returned to the background $DMS_a$ level as the WS increased from 3.4 to 5.8 m s$^{-1}$ (Fig. 3c). According to back trajectories at that time the source of this long lasting

maximum $DMS_a$ spike in the wet season dataset is suspected to have originated from air that traversed the Capricorn Bunker Group, an extensive array of coral reefs to the SE of Heron Island that span a distance of ~70 km (Fig. 1b). This observation supports a previous report of elevated $DMS_a$ in SE trade winds that travelled over dense coral biomass areas of the GBR and western Pacific Ocean (Jones and Trevena, 2005). Additionally, at a Queensland coastal site (24.21°S, 151.90°E), elevated new particle number concentrations, reflecting a strong nucleation event, were detected in an air stream that travelled over the

Bunker Group of southern GBR reefs on 30 March 2007 (Modini et al., 2009).

### 3.2 Dry season 2013

During the dry season campaign at Heron Island, *spring* tides were experienced from 18 to 24 July with low tide heights ranging from -0.1 to 0.3 m. During these particularly low tides ten clearly defined $DMS_a$ spikes of varying magnitude were observed above the background oceanic $DMS_a$ signal. Three of these spikes were detected in WS exceeding 5 m s$^{-1}$ which

suggests that they were intense emission events from the lagoonal platform reef. An extreme $DMS_a$ spike was detected in the early evening of 25 July when a convection storm deposited only 1.8 mm of rainfall onto the Heron Island reef flat when much of it was aerially exposed by a *spring* low tide of 0.2 m. Under this scenario, the background $DMS_a$ concentration of 1.4 nmol m$^{-3}$ at 17:26 increased to 45.9 nmol m$^{-3}$ (1122 ppt) by 17:50 (Figs. 6a & 6b). This rain-reef-atmosphere interaction was characterized by a strong odour of DMS, and to our knowledge is the highest $DMS_a$ concentration measured over a coral

reef. It was also the sharpest $DMS_a$ spike detected during both campaigns; the $DMS_a$ returning to the background level within 1 h. This 45.9 nmol m$^{-3}$ $DMS_a$ peak concentration occurred under a low rain rate (1.5 mm h$^{-1}$) and one of the highest WS (9.5 m s$^{-1}$) measured during the dry season campaign. At this WS, rain effects on air-water gas exchange are minor (Harrison et al., 2012) and cannot account for the sudden intensity of $DMS_a$ observed. Since physical reasons alone cannot explain this extreme $DMS_a$ spike, we propose that rainfall onto the aerially exposed coral reef induced a biological shock, where the coral

reacted to a rapid decline in seawater salinity by utilizing intracellular DMSP to maintain osmotic pressure balance and to cope with the associated rapid decline in temperature (Stefels, 2000, and references therein). DMSP is recognized as an osmotically active intracellular compatible solute in unicellular algae that may act to buffer cell volume changes during the initial period after an osmotic shock (Kirst, 1996). Since unicellular algae (zooxanthellae) are abundant within the coral



symbiosis, it is likely that corals use DMSP for osmotic regulation. Utilization of DMSP, a sulfonium zwitterion, for osmotic pressure balance could generate significant quantities of DMS as a metabolic by-product resulting in the extreme $DMS_a$ spike observed, even under strong atmospheric mixing. It is suspected that the brevity of this $DMS_a$ spike was because DMS was directly exchanged to the atmosphere from aerially exposed coral on the very low tide, and was rapidly dispersed by the

strong winds.

### 3.3 Estimation of seasonal DMS flux

During both campaigns at Heron Island, daily backward trajectory air parcel analysis showed that clean marine air flows were received the majority of the time. Occasionally, when air was received from over the Australian continent, it was derived from high altitude (+1000 m), and was indicated to be clean free-tropospheric air. These conditions support use of

the applied photochemical ambient mass balance Eq. (1) which specifies that $DMS_a$ is predominantly removed from the clean MBL by reaction with OH. In polluted regions, where significant concentrations of $NO_3$ are often present, it is necessary to include $NO_3$, another DMS oxidant, in the mass balance equation used to determine DMS flux (Chen et al., 1999; Shon et al., 2005). Equation (1) balances factors that alter mean concentrations of $DMS_a$ over time within seasonal average MBL box volumes. The MBL is composed of the surface layer, immediately above the sea, and the mixed layer

(sub-cloud layer); it is through these layers that the ocean and the atmosphere are coupled. The mixed layer is capped by a transition layer that is ~100 to 200 m deep over tropical oceans (Johnson et al., 2001). This zone, often referred to as the entrainment zone, is characterised by a decrease in humidity together with a sharp increase in stability leading into the free troposphere (Clarke et al., 1998). We were unable to measure all of the input variables for Eq. (1) so representative values for the study location were derived from the literature. A value of 0.004 m s$^{-1}$ was applied for $E_v$ according to average data

obtained from Lagrangian experiments in the southern hemisphere remote MBL (Wang et al., 1999). A value of 6.5 x 10$^{-12}$ cm molecule$^{-1}$ s$^{-1}$ was applied for $K$, which is the sum of the abstraction and addition rate reactions of OH with DMS at standard temperature and pressure (Finlayson-Pitts and Pitts Jr, 2000). The major variable altering OH concentrations in the atmosphere is the intensity of solar UV-B radiation, which controls the generation of OH from the photolysis of ozone (Rohrer and Berresheim, 2006). At Heron Island, the average daily solar irradiance measured at the surface during the 2013

dry season was only 10 % lower than during the 2012 wet season campaign (Table 1). According to this seasonal variation in surface solar irradiance and reported average OH concentrations over the South Pacific Ocean (Seinfeld and Pandis, 1998), values for [$OH$] of 1.8 x 10$^6$ (wet season) and 1.6 x 10$^6$ molecules cm$^{-3}$ (dry season) were applied to Eq. (1). Average mixed layer depths (MLD) at noon were 977 m (range 680 to 1460 m, $n$ = 15 days) and 786 m (range 346 to 1312 m, $n$ = 19 days) during the 2012 wet season and 2013 dry season campaigns, respectively. These average values are consistent with a study

of the MLD at Heron Island in June 2009 and February 2010, which ranged from 375 to 1200 m above the surface (MacKellar et al., 2013). Deepest MLDs were observed under stable anti-cyclone conditions, while shallowest MLDs were observed during periods of heavy precipitation with convective downdrafts. MLDs often coincide with the MBL height, which is typically around 700 to 800 m (Stull, 1988); the average MLD values of 977 m (wet season) and 786 m (dry season)





were, therefore, applied for $H$ in Eq. (1). Mean concentrations of $DMS_a$ measured in this study were 1.3 (1σ = 1.6, $n$ = 923) and 3.9 nmol m$^{-3}$ (1σ = 1.5, $n$ = 651) for the dry and wet seasons, respectively (Table 1). The number of $DMS_a$ measurements is sufficiently large that these mean concentrations for each campaign are expected to be representative of $DMS_a$ in the MBL over Heron Island during the wet and dry seasons. The concentration of $DMS_a$ in the entrainment zone is

5 reported to be typically ~10 % of the MBL concentration (Ayers et al., 1995; Chen et al., 1999) so this percentage was applied to determine [$DMS_t$]. Solving Eq. (1) for $F_{DMS}$ using the mean $DMS_a$ concentrations and other specified values, gave surface fluxes of 5.0 and 1.4 µmol m$^{-2}$ d$^{-1}$ for the wet and dry season campaigns, respectively. These fluxes were clearly dominated by the oceanic background $DMS_a$ source; they reflect expected seasonal surface ocean primary productivity, and are temporally and spatially consistent with predicted fluxes calculated from a database of global surface ocean DMS

concentrations (Lana et al., 2011). These seasonal fluxes, calculated using the photochemical mass balance approach, are expected to have a relative uncertainty of ~50 % (Chen et al., 1999).

The highest $DMS_a$ concentrations in both seasonal datasets could be attributed to coral reef emissions. This was most apparent in the dry season dataset (Fig. 6), where the relatively stable oceanic background $DMS_a$ signal was occasionally elevated by spikes of $DMS_a$ from the Heron Island reef flat at low tide. In both seasons the reef spikes positively skewed the

15 dataset distributions resulting in larger mean values than median values. If median $DMS_a$ concentrations (Table 1) are instead entered into Eq. (1), $F_{DMS}$ equates to 4.8 and 1.2 µmol m$^{-2}$ d$^{-1}$ for the wet and dry season campaigns, respectively. Entering median $DMS_a$, concentrations into Eq. (1) provides an estimate of the oceanic flux in both seasons since the median values act to negate the contribution from the coral reef $DMS_a$ spikes. The difference between $F_{DMS}$ calculated using mean and median values for [$DMS$] equates to 0.2 µmol m$^{-2}$ d$^{-1}$ for both the wet and dry season campaigns, thereby providing an

20 estimate of the previously unquantified contribution of the coral reef to $F_{DMS}$ at Heron Island. When this 0.2 µmol m$^{-2}$ d$^{-1}$ coral reef flux estimate is expressed as a fraction of the overall $F_{DMS}$ in each season, it is apparent that the coral reef played a significantly greater contribution during the dry season. The coral reef enhanced the dominant oceanic $F_{DMS}$ by 4 % during the wet season and 14 % during the dry season campaign, where the dry season flux enhancement resulted largely from the 45.9 nmol m$^{-3}$ $DMS_a$ spike generated by a rain rate of only 1.5 mm h$^{-1}$ onto the aerially exposed coral reef at low tide.

Clearly, considerable uncertainty is inherent in the estimated coral reef DMS flux due to the difficulty of distinguishing the coral reef and oceanic $DMS_a$ source contributions; however, it provides a starting point for future improvement.

## 4 Conclusions

This study has provided environmental evidence that coral reefs in the vicinity of Heron Island are point sources of $DMS_a$, where emissions may at times be detectable as spikes of $DMS_a$ above the background oceanic signal. Detection of these

30 $DMS_a$ spikes relies on extended continuous measurements with sufficient frequency to resolve the spikes from the background source signal. The automated GC used here has served that purpose; however, further on-site continuous sampling of $DMS_a$ at the GBR is required to more closely examine factors that cause coral reefs to emit DMS to the atmosphere. For example, the higher temporal resolution possible with proton transfer reaction mass spectrometry (Lawson et



al., 2011) and atmospheric pressure chemical ionisation mass spectrometry (Bell et al., 2013) may provide additional insights into those factors and their interaction, while also improving the estimation of $F_{DMS}$ from the GBR. We found the ocean to be the dominant source of $DMS_a$ at Heron Island, where the ocean source was supplemented by occasional coral reef-derived spikes of $DMS_a$ that were highly variable irregular events generally occurring at low tide when conditions exist that can stress the reef. The extreme $DMS_a$ reef spike recorded in the dry season of 2013, which we conclude was an acute biological response to suddenly unfavourable environmental conditions, demonstrates that the Heron Island lagoonal platform reef has a unique DMS emission mechanism when compared with the surface ocean where wind driven turbulence largely controls the air-sea exchange of DMS. Furthermore, the extreme $DMS_a$ spike in the winter dry season demonstrates that the Heron Island lagoonal platform reef can be a seasonally aberrant source of $DMS_a$ in comparison to the surface ocean where increased primary production in the summer provides consistently higher concentrations of $DMS_a$ than the dormant winter period (Ayers et al., 1991). The seasonal aberration of the coral reef as a source of $DMS_a$ is supported by the flux estimates, where the coral reef enhanced the dominant oceanic $F_{DMS}$ by 14 % during the dry season but only 4 % during the wet season campaign. Another significant finding from this study is that convective precipitation intensified the low tide emission of DMS from the coral reef surrounding Heron Island. On a broader regional scale this process may contribute to sulfate-derived secondary aerosol that may ultimately influence the radiation budget over the GBR; however, the extent to which $DMS_a$ contributes to aerosol production and its role in CCN formation over the GBR is currently unknown. Aerosol formation and evolution studies are, therefore, required to determine if the GBR is a climatically important source of marine aerosol.

*Author contributions.* H.B. Swan configured the instrumentation, collected, processed and interpreted the data, and wrote the manuscript, gaining edits and textual contributions from the co-authors. G.B. Jones initiated the field study program and, together with E.S.M. Deschaseaux, participated in some of the field work to assist data collection. All authors approved the study design.

*Acknowledgements.* This research was funded from grants to G.B. Jones by the Marine Ecology Research Centre of Southern Cross University (SCU), the Australian Institute for Marine Science (AIMS, Townsville, Qld), and Australian Research Council Discovery Grants DP110103638 and DP160100248 awarded to B.D. Eyre. H.B. Swan would like to thank George Tannous for constructing the autosampler, the National Measurement Institute Australia for providing some of the equipment used to collect the $DMS_a$ datasets, Edith Swan for assistance with preparation of graphics, and John Ivey for helpful comments on the manuscript. All authors would like to thank the Heron Island Research Station staff and management for assisting with our experimental needs, and Melissa MacKeller (University of Queensland) for providing Fig. 1b. We gratefully acknowledge the NOAA Air Resources Laboratory (ARL) for provision of the HYSPLIT transport and dispersion model and READY website (http://www.ready.noaa.gov) used to obtain back trajectories and mixed layer depths.



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

**Table 1. Summary of atmospheric DMS and some meteorological data for the 2012 wet season and 2013 dry season at Heron Island, southern Great Barrier Reef**

| Season | Atmospheric DMS (nmol m$^{-3}$) | | Wind speed (m s$^{-1}$) | | Air temp (°C) | | Water vapour mixing ratio (g kg$^{-1}$) | | Daily solar irradiance[a] (MJ m$^{-2}$ d$^{-1}$) | |
|---|---|---|---|---|---|---|---|---|---|---|
| | Wet | Dry | Wet | Dry | Wet | Dry | Wet | Dry | Wet | Dry |
| Mean | 3.9[b] | 1.3[c] | 4.5[d] | 3.5[e] | 26.5 | 20.8 | 17.3 | 8.9 | 27.4 | 24.7 |
| SD | 1.5 | 1.6 | 2.4 | 2.3 | 1.7 | 2.2 | 1.6 | 2.2 | 10.6 | 8.6 |
| Median | 3.7 | 1.2 | 4.1 | 3.3 | 26.0 | 20.4 | 17.6 | 9.2 | 32.0 | 28.3 |
| Minimum | 0.6 | 0.2 | 0 | 0 | 22.7 | 15.6 | 12.8 | 2.2 | 4.7 | 5.0 |
| Maximum | 11.5 | 45.9 | 13.6 | 9.9 | 32.6 | 29.0 | 20.8 | 13.2 | 37.2 | 33.2 |

[a]Daily solar irradiance is for the 24 h period.
[b]The number of DMS measurements for the 2012 wet season campaign was 651.
[c]The number of DMS measurements for the 2013 dry season campaign was 923.
[d]The number of meteorological observations for the 2012 wet season campaign was 1313.
[e]The number of meteorological observations for the 2013 dry season campaign was 1716.



**(a)**

**(b)**

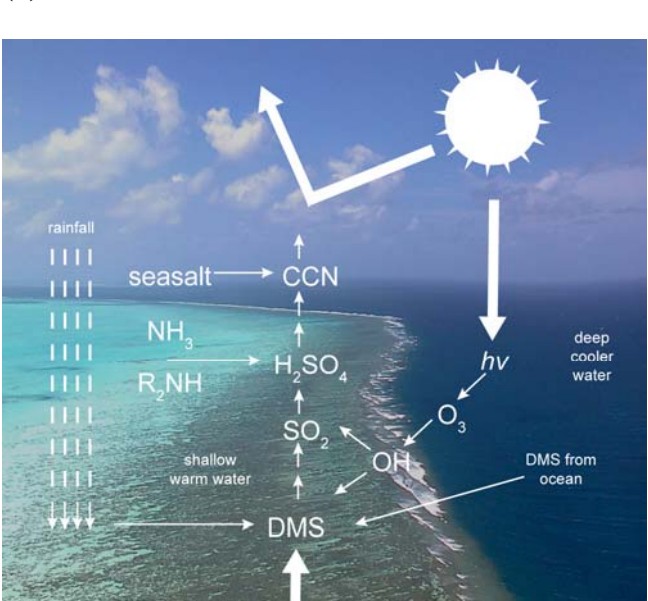

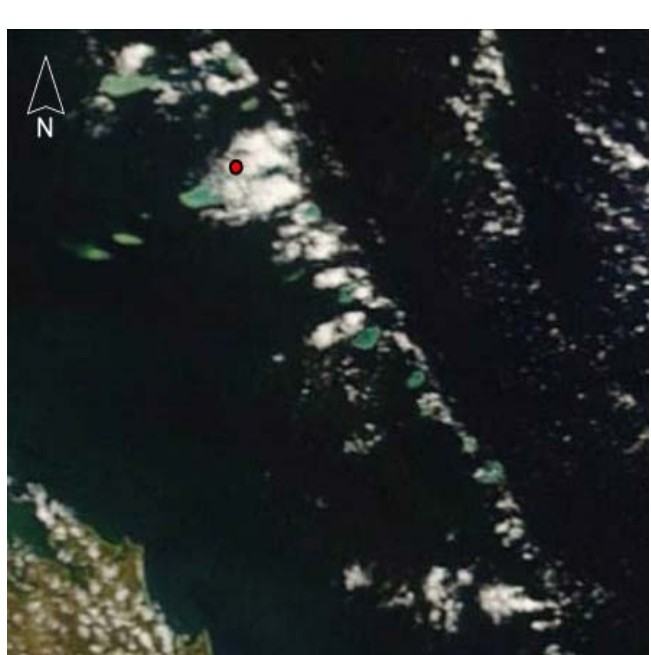

**Figure 1: The Capricorn Bunker Group of coral reefs, southern Great Barrier Reef, Australia (23.13°S, 151.85°E to 23.92°S, 152.60°E). (a), an aerial photo of Wistari Reef near Heron Island (Image: H.B. Swan). Superimposed on this image is a conceptual model of factors controlling DMS$_a$ derived sulfate aerosol production over the GBR. Ocean derived DMS$_a$ and spikes of DMS$_a$ from the coral reef at low tide are oxidized by photochemically produced hydroxyl radical (OH) forming sulfate aerosol that can grow to cloud condensation nuclei (CCN). This process can assist formation of high-albedo low-level marine clouds that influence the radiation budget of the GBR. (b), MODIS (Moderate Resolution Imaging Spectroradiometer) satellite image of daytime low-level convective clouds aligned over the Capricorn Bunker Group of coral reefs. Heron Island, near Wistari Reef (cloud covered), is indicated by a red circle. Part of the east Australian coastline is seen in the bottom left-hand corner.**



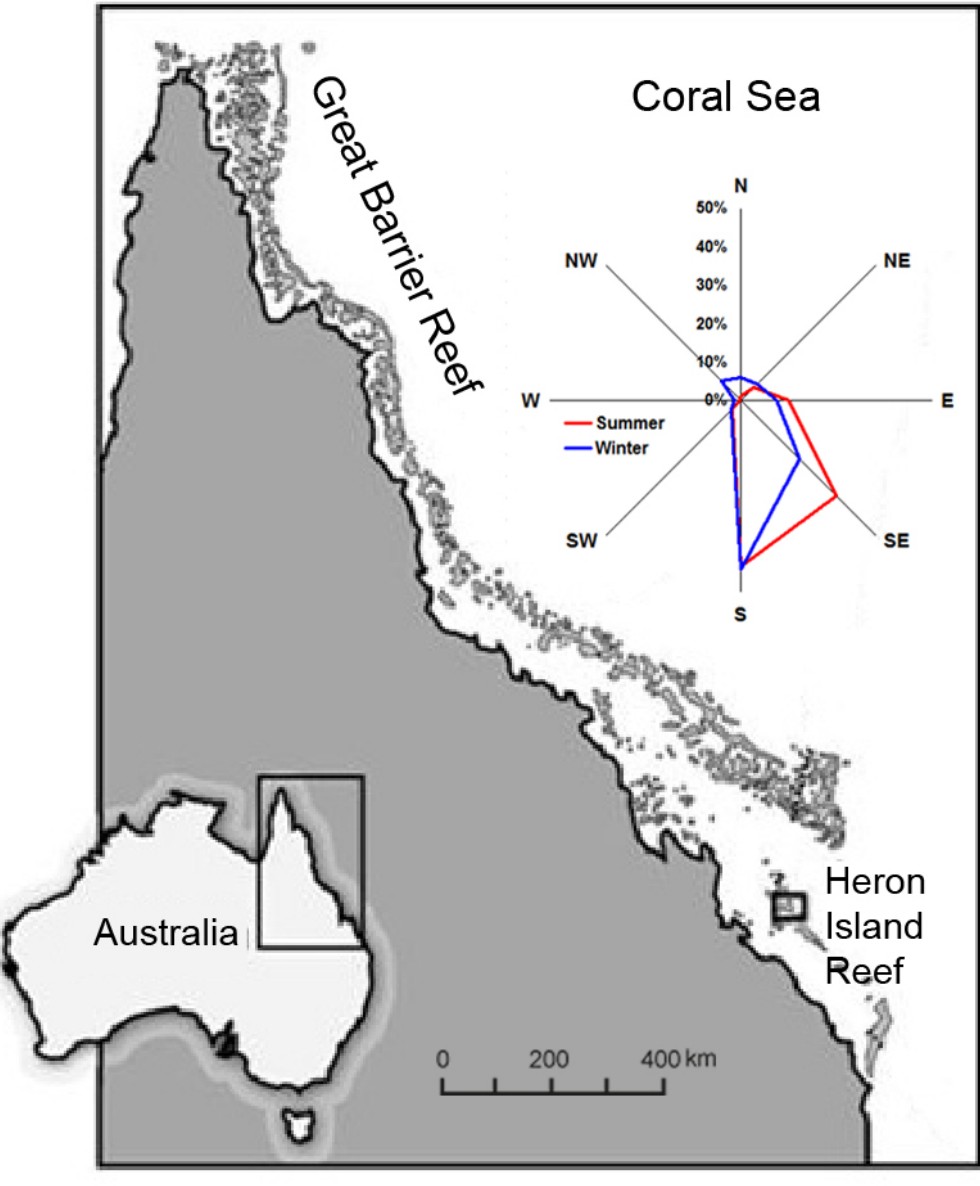

**Figure 2: Location of Heron Island in the southern Great Barrier Reef, Australia, where continuous on-site analysis of DMS$_a$ was conducted in the austral wet season of 2012 and dry season of 2013. The compass shows the directional frequency of winds received at Heron Island during the late summer wet season (red line) and mid winter dry season (blue line) measurement campaigns. Daily backward trajectory analysis showed that marine-derived air streams in the SE sector were received at Heron Island the majority of the time during both campaigns.**



**(a)**

**(b)**                                          **(c)**

**Figure 3: Data obtained from Heron Island during the wet season of 2012. (a), Entire time series of DMS$_a$ (red line), wind speed (WS, grey line), rainfall (green line) and tide height (blue line). Four distinct spikes (10, 14, 16-17 March) above the background DMS$_a$ signal are indicated to be coral reef-derived emissions. (b), Extracted time series showing the DMS$_a$ spike on 10 March (red line) which occurred during a 0.3 m low tide under still conditions (grey line). There was no rainfall during the period shown. (c), Extracted time series showing three distinct coral reef DMS$_a$ spikes. The first, detected between 21:12 on 14 March and 1:05 on 15 March (~4 h duration), was associated with convective rainfall (green line) during a 0.8 m low tide. The highest DMS$_a$**



concentration recorded during the wet season campaign (11.5 nmol m$^{-3}$) was detected on 16 March at 6:27, shortly after high tide. This tidally unique and longest lasting DMS$_a$ spike (~8.5 h) is indicated to have originated from low-level marine air that traversed the length of the 70 km Bunker Group of coral reefs that lie to the SE of Heron Island (Fig. 1b). The third DMS$_a$ spike seen on 17 March, which occurred on a 0.8 m low tide under abating WS, is indicated to be derived from the lagoonal platform reef surrounding Heron Island.





**Figure 4: Twelve hour separated back trajectories of 48 hour duration ending 10 March 2012 showing air movement onto the Australian continent with high altitude free-tropospheric air (red line) being directed to Heron Island. This relatively dry non-marine air stream resulted in the lowest $DMS_a$ concentration of 0.6 nmol m$^{-3}$ during the wet season campaign, which occurred shortly after high tide. However, by low tide a $DMS_a$ spike peaking at 6.4 nmol m$^{-3}$ was detected in this air stream under calm conditions (Fig. 3b) prior to a southerly change that redirected the usual marine air flow. The tidal and meteorological conditions under which this $DMS_a$ spike was detected strongly suggest that it was derived from the coral reef surrounding Heron Island.**







**Figure 5: Twelve hour separated back trajectories of 48 hour duration ending 16 March 2012 showing the low-level SE marine air flow (red line) that is indicated to have traversed the ~70 km length of Capricorn Bunker Group of coral reefs (Fig. 1b) before arriving at Heron Island. This marine air flow resulted in the longest lasting DMS$_a$ spike and the highest recorded DMS$_a$ concentration of 11.5 nmol m$^{-3}$ during the wet season campaign (Fig. 3c).**



**(a)**



**(b)**

**(c)**

**Figure 6: Data obtained from Heron Island during the dry season of 2013. (a), Entire time series of DMS$_a$ (red line), WS (grey line), rainfall (green line) and tide height (blue line). The second axis for DMS$_a$ is scaled to show the relative intensity of the coral reef DMS$_a$ spike (45.9 nmol m$^{-3}$) detected at low tide in the early evening of 25 July. Even under relatively strong atmospheric mixing (WS = 9.5 m s$^{-1}$, grey line) this DMS$_a$ reef-derived spike was detected as an intense event. (b), Extracted time series from 20 to 26 July showing ten low tide coupled DMS$_a$ spikes from the Heron Island reef flat. The extreme DMS$_a$ spike (off-scale) on 25 July can be seen to be linked to the 0.2 m low tide and a brief shower of rain that fell onto the aerially exposed coral reef. (c), Time-line magnification of the low tide induced DMS$_a$ spike from the Heron Island coral reef on 23 July between 13:31 and 17:28. There was no rainfall during the period shown.**