# Peer review of "Manuscript bg-2016-387"

_Biogeosciences, 2016_

## Referee Comment (RC1) · Anonymous Referee #1 · 31 Oct 2016

The paper by Swan et al., "Coral reef originals of atmospheric dimethylsulfide at Heron Island, southern Great Barrier Reef, Australia" is an analysis of the contribution of corals to DMS atmospheric loading. It is well known that the microbial community in the ocean is an important source of DMS to the marine atmsophere. The same types of physiological processes that produce DMS in the microbial organisms also occurs in corals, but the contribution of corals to atmospheric DMS is less well-known. The authors discuss that the areal extent of coral reefs might be small, but the size of the living organism is large and, therefore, the potential emissions of DMS might be important. A field study during two different seasons illustrates the magnitude and enivronments factors influencing DMS emissions from corals around Heron Island. There is a clear

indication that the corals are important emitters of DMS, especially during low tide and when stressed. The need to understand the larger area, as well as other coral reefs in the world oceans, is evident. The manuscript is a useful contribution to the general knowledge on DMS cycling in the Earth System. It is well written and the conclusions are largely well drawn from the evidence presented. I think the manuscript should be accepted after the minor revisions, described below, are addressed.

Specific Commnents: 1) Figure 1b is really a great figure and compelling for the point of the paper, but not discussed at all. 2) The wind compass in Figure 2 is a bit confusing. It would be easier to read and comprehend if it was in a wind rose format. 3) The description of the GC sampling could be more detailed. I see that the authors cited a previous methods paper, I would still like to know more about this particular experiment. What size tubing? How fast was the sample pumped? How was it pumped? Why was the air trapped 14 mins? 4) I don't like the units of nmol/m3 and appreciate when ppt units also mentioned. Can this be done throughout? 5) Style of the discussion sections seems a bit off - I miss a general description of the results. Instead, the manuscript seems to delve right into the spikes and the effect of tides. 6) How do the authors know that the large spike found during the dry season (shown in figure 6a) is not an instrumental problem? Why should there only be biological shock during the dry season (it seems that there are periods during the wet season also without rainfall and with low tide)? 7) Why didn't the authors play a bit with their results, for example looking at forward trajectories to see where the DMS ends or scaling up to the entire GBR? 8) How was H (or MLD??) actually determined? And why do the authors call it MLD? This is terminology used more for water mixed layers. Why not always say H? Or are these values somehow different?

---

## Referee Comment (RC2) · Anonymous Referee #2 · 1 Nov 2016

The paper "Coral reefs origins of atmospheric dimethylsulfide at Heron Island, southern Great Barrier Reef, Australia" by Swan et al. is an interesting study that helps to understand if and under which environmental conditions coral reefs seems to contribute to atmospheric DMS emissions. Due to continuous DMS measurements during two different seasons, the study gives a great overview about how atmospheric DMS loading by coral reefs changes with time and which environmental conditions influenced the emissions. The paper is well structured and written, however, minor revisions are necessary.

Method: The description of the measurement procedure is insufficient. More details are necessary to understand how atmospheric DMS was measured, e.g. what is the

cryogenic trap consist of, is the DMS preconcentrated and trapped before analysis?

Is it right that the inlet of the measurement device is around 100 m away from the coral reef? If is it so how you can be sure that the DMSa you measured is directly emitted by the reef.

Give more details about how you determined low and high tides. You give even a negative value (p6 line 18). Did you use the height of the reefs as a zero-point?

In section 2.2 "Flux calculation" you introduced the mass balance equation. Did you perform an error estimation of the different parameter of the equation? Did you estimate the variability of the parameter over time? A mass balance calculation can exhibit many errors due to uncertainties of the different parameter and their variability over time. You have to discuss in more detail that the different parameters you are used are reasonable.

Results and Discussion: An overall description of your data is missing. What are the general patterns of your data? Is there a general trend? Additionally, you start directly with the interpretation of the peaks without any introducing sentences. Say in the beginning shorty what you have done and why and what you found.

In the first paragraph (p5 l3-10) you mentioned many time points which is hard for the reader to follow. Additionally, the different time points are hard to see in fig. 3. Maybe show clearly in the fig the time steps you described in detail and maybe reword the text a little bit for a better understanding for the reader.

P6 L3-15: Why you talked in this paragraph about the measurements on 16 March and before about data from 17 March. Why it is not in chronological order?

The mixed layer depth (MLD) you mentioned in the text (p7 line 28) is it in the water or in the atmosphere. Is it the same like the MBL? The MLD is generally used for the water. Please clarify.

Can you discuss in the results and discussion section the stress level and health conditions of the coral reef you investigated? Is the reef already affected by global change (temperature, pH), has it a high biodiversity, was coral bleaching observed? Can these factors affect the DMSP and DMS production? Are the events observed during the measurements (very low tides, reef exposure to the air, rainfall on the corals) normal events which occurred on a regular base or were these extreme and seldom events?

It would be also interesting to measure directly DMS emissions by the corals in incubation experiments under different environmental conditions to have the direct evidence that the DMSa is coming from the corals directly. It is clear that this cannot be part of this study but is interesting to investigate in future studies.

Figures: Fig 1 is not necessary to understand the paper and is not discussed in detail in the paper. It has not important new information. I recommend to delete it.

Figures 3 and 6 are hard to read. The grey, green and blue colors are hard to distinguish and there are too many parameters in one graph. Additionally, you discussed a lot time points but they are hard to see in the sub-panels. See comment above.

---

## Author Response (AR1)

**Manuscript bg-2016-387**

**Replies to Referee #1**

*Comment 1:*
*Figure 1b is really a great figure and compelling for the point of the paper, but not discussed at all.*

Reply:
Fig. 1b is mentioned on P6, L11 to pictorially describe the Capricorn Bunker Group of coral reefs to the SE of Heron Island. Fig. 1b was not discussed in the context of low-level cloud formation over these coral reefs because it is a contentious image among atmospheric scientists. While there may be broad agreement that convective processes have contributed to cloud alignment over the Capricorn Bunker Group of coral reefs shown in the MODIS image, we provide insufficient evidence to conclude that atmospheric DMS ($DMS_a$) derived from those reefs is the source of those clouds. In our manuscript we provide evidence that the Heron Island reef flat and the Capricorn Bunker Group of reefs can at times be significant sources of $DMS_a$. Those spikes of $DMS_a$ can potentially contribute to development of clouds over the GBR; however, as stated in the final sentences of the conclusion "the extent to which $DMS_a$ contributes to aerosol production and its role in CCN formation over the GBR is currently unknown. Aerosol formation and evolution studies are, therefore, required to determine if the GBR is a climatically important source of marine aerosol." This is the crux of the matter now requiring further investigation. After reading our manuscript we would like the reader to draw their own conclusion regarding the cloud cover over the reefs shown in Fig. 1b.

Another referee of the manuscript has recommended deleting Fig 1, stating the figure is not necessary to understand the information in the paper, and that it does not provide important new information. This contrasting comment possibly alludes to the contentious nature of the images shown in Fig 1. We wish to promote discussion by presenting this figure, but do not want to draw conclusions beyond what the data we collected allows. The conclusions we have made are in line with the objective of the study stated in the final sentence of the introduction.

*Comment 2:*
*The wind compass in Figure 2 is a bit confusing. It would be easier to read and comprehend if it was in a wind rose format.*

Reply:
The radar compass plot was used in Fig 2 because it provided the clearest representation of the frequency of wind directions for the two seasons when plotted together. When the wind direction data for both seasons is plotted on a wind rose style plot, it is difficult to distinguish the seasonal differences because they were minor. What we want to convey by including the compass plot in Fig 2 is to show that there was little difference in the directional frequency of the winds at Heron Island in both the wet and dry seasons.

*Comment 3:*
*The description of the GC sampling could be more detailed. I see that the authors cited a previous methods paper, I would still like to know more about this particular experiment. What size tubing? How fast was the sample pumped? How was it pumped? Why was the air trapped 14 mins?*

Reply:
In order to minimise the length of Section 2: 'Methods', a reference is provided to a 2015 publication that gives a complete description of the instrumentation and its measurement uncertainty. We appreciate that the referee would like more methodological information provided in this results focussed paper, so it will be included in the revision. Answers to the particular questions posed are as follows: Marine air was drawn through 6 mm internal diameter Teflon tubing via a high-capacity oxidant scrubber composed of 1% w/v sodium ascorbate and glycerol impregnated into a 47 mm diameter glass-fibre filter. Surface-level marine air was drawn through the sampling system at a flow rate of approximately 260 mL min$^{-1}$ using a single-stage diaphragm vacuum pump (Vacuubrand, model ME2, Germany). The air was drawn into a cryogenically

cooled trap (cryotrap) that was constructed by passing 1.6 mm diameter Teflon tubing through ~50 cm of copper tubing of 2.0 mm internal diameter, and bending it into a loop. A sample collection time of 14.4 min was used to deliver 3.72 L of air into the cyrotrap. This volume of air was required to concentrate sufficient $DMS_a$ for chromatographic analysis to provide a 0.1 nmol m$^{-3}$ (2 ppt) reporting limit. Calibration was achieved using permeated ethyl methyl sulfide.

*Comment 4:*
*I don't like the units of nmol/m3 and appreciate when ppt units also mentioned. Can this be done throughout?*

Reply:
The photochemical ambient mass balance Eq. 1 used to determine seasonal DMS emission flux requires that $DMS_a$ molar concentrations are entered to determine flux in the usual units of $\mu$mol m$^{-2}$ d$^{-1}$. This is the main reason why $DMS_a$ molar concentrations are reported throughout the manuscript. We are aware that $DMS_a$ concentrations are dependent on pressure and temperature according to the ideal gas law, and that mixing ratios are more suitable than concentrations to describe the abundance of species in air, particularly when samples are collected in aircraft over various altitudes. All the $DMS_a$ measurements we made were at sea level under relatively consistent temperature and pressure. Additionally, the $DMS_a$ concentration of nmol m$^{-3}$ used throughout the manuscript is a SI unit. In contrast, ppt is not an SI unit, and it is recommended by IUPAC that the SI unit of pmol mol$^{-1}$ is used in place of ppt notation. However, we understand that it is customary for trace species in air to be reported in ppt because familiarity with this unit allows many readers to readily compare abundances of trace gaseous species. We would like to present SI concentrations throughout the manuscript by reporting our surface $DMS_a$ measurements in nmol m$^{-3}$ to maintain consistency with emission fluxes reported in $\mu$mol m$^{-2}$ d$^{-1}$. Nevertheless, Table 1 will be edited to include the corresponding ppt mixing ratios for the $DMS_a$ concentrations given in the table. The limit of reporting of 2 ppt will also be given in brackets next to the concentration in Section 2.1 (P3, L17); however, we think it will clutter the manuscript text to report ppt mixing ratios in brackets next to every concentration mentioned in Section 3: 'Results and Discussion'.

*Comment 5:*
*Style of the discussion sections seems a bit off - I miss a general description of the results. Instead, the manuscript seems to delve right into the spikes and the effect of tides.*

Reply:
An introductory paragraph that provides a general description and summary of the results will be included at the beginning of Section 3: 'Results and Discussion' in the revised manuscript. This introductory paragraph will not repeat information given in Section 4: 'Conclusions'.

*Comment 6:*
*How do the authors know that the large spike found during the dry season (shown in figure 6a) is not an instrumental problem? Why should there only be biological shock during the dry season (it seems that there are periods during the wet season also without rainfall and with low tide)?*

Reply:
The intense $DMS_a$ spike detected during the dry season campaign was not an instrumental problem. As stated in the manuscript, there was an intense odour of DMS at the time the spike was detected, which was evident to a number of people who remarked about the unusually strong "marine odour" coming from the platform reef surrounding the island. Although the $DMS_a$ spike was relatively sharp it was not derived from a single measurement. The oceanic background $DMS_a$ concentration on 25 July 2013 ranged from 1.0-1.8 nmol m$^{-3}$ over the entire day prior to the intense $DMS_a$ spike in the early evening at 17:50 when the $DMS_a$ rapidly rose to 45.9 nmol m$^{-3}$. The following $DMS_a$ measurement was 9.7 nmol m$^{-3}$ at 18:14, which tapered off from 2.7 nmol m$^{-3}$ at 18:38 to the preceding background concentration of 1.5 nmol m$^{-3}$ by 21:24. As mentioned in the discussion, this $DMS_a$ spike was brief due to rapid dilution by strong horizontal advection under a wind speed of 9.5 m s$^{-1}$ at the time of the spike. These conditions require higher resolution sampling to adequately capture this sort

of DMS$_a$ reef emission, which is one of the reasons why it is recommended in the conclusion that chemical ionisation mass spectrometry could assist further studies of DMS$_a$ emissions from the GBR.

We are not saying in the manuscript that biological shock to the reef will only occur during the dry season. It just happened to be during the 2013 dry season campaign that we detected the particularly intense DMS emission that appeared to be brought about by coincidence of environmental conditions that were unfavourable to the coral reef. These conditions included *spring* low tides which aerially exposes more of the reef for longer periods than neap low tides, coinciding with a brief shower of rain onto the reef flat at the time when the reef was most exposed. Such conditions could occur at other times of the year. *Spring* low tides occur during December-January in the wet season, so it is possible that intense emissions of DMS could occur at that time if the exposed reef was showered by rainfall. In a previous study on the GBR in February (austral summer) elevated seawater DMS concentrations (54 nM) were measured during low tide when there was 20 minutes of rainfall, which reduced the seawater salinity by 0.75 PSU (Jones et al., 2007, Environmental Chemistry, doi:10.1071/EN06065). This situation is likely to have released a wet season DMS$_a$ spike. As is evident from the entire winter dataset we show here, the intensity of the DMS$_a$ spike detected in the early evening of 25 July 2013 was a unique event, and it was good fortune to be on-site at that time with equipment to detect and quantify it.

The referee comments that there are periods during the wet season also without rainfall and with low tide. Fig. 3a shows a period between 8 March and 17 March 2012 when there was only a few convective derived short showers. Low tide during periods of dry weather resulted in few detectable DMS$_a$ spikes from the coral reef. In contrast, the convective shower that coincided with low tide in the evening of 14 March produced the second largest DMS$_a$ spike (10.6 nmol m$^{-3}$) recorded during the late summer wet season campaign. This indicates that rain on the reef in the summer season can result in a DMS$_a$ spike similar to the spike detected during the winter campaign. The reason that the DMS$_a$ spike during the winter was much more intense than the one detected during the summer is not clear, but it may have to do with factors such as the level of the low tide, the time that the coral had been aerially exposed, the temperature of rain that fell on the reef and the resulting surface seawater temperature. It was apparent that it was not the amount of rainfall but *when* the rainfall occurred that led to detectable DMS$_a$ spikes from the coral reef.

*Comment 7:*
*Why didn't the authors play a bit with their results, for example looking at forward trajectories to see where the DMS ends or scaling up to the entire GBR?*

Reply:
A few forward trajectories were made to observe air mass transport away from Heron Island. The reason these forward trajectories were prepared was to examine where Wedge-tailed Shearwaters (Mutton Birds) that nest on Heron Island might be going to find food. The forward trajectories indicated that if they flew with the wind they would often travel to the Swain Reefs, an extensive reef system to the north of Heron Island (seen in Fig. 2). If this forward trajectory analysis is extended to transport of DMS$_a$ spikes from Heron reef it is possible that the DMS$_a$ oxidation products might contribute to CCN formation over the Swain Reefs. We do not present evidence in the manuscript to support this possibility and the lead author is reluctant to extrapolate or scale up our results obtained at Heron Island on the southern GBR to the entire 2,300 km length of GBR. The GBR stretches 13 degrees of latitude along the Queensland coastline and it is risky to assume that processes operating in the northern GBR are the same as those on the southern GBR. The stark contrast in the extent of coral bleaching on the northern GBR compared to the minimal bleaching on the southern GBR during the summer of 2015-16 is a recent pertinent example of differences that can occur along the length of the GBR. Our manuscript was previously submitted to another journal and it was criticised by each referee for the suggestion that the coral reef DMS$_a$ spikes observed at Heron Island on the southern GBR may occur over the entire GBR leading to formation of low-level marine clouds that possibly constitutes a regional climate feedback. At the present time that hypothesis remains speculative and much more research is required to determine if the GBR is a climatically influential source of marine aerosol, as stated in the last sentence of the conclusion.

*How was H (or MLD??) actually determined? And why do the authors call it MLD? This is terminology used more for water mixed layers. Why not always say H? Or are these values somehow different?*

5   Reply:
Atmospheric MLDs were obtained using the Hybrid Single-Particle Lagrangian Integrated Trajectory (HYSPLIT) transport and dispersion model developed by NOAA's Air Resources Laboratory, as stated in Section 2.2 of the manuscript. The model, which is described by Stein et al (2015), uses sophisticated computations of atmospheric transport and mixing.

10  The referee may be more familiar with the MLD when used in the marine context; however, meteorologists and atmospheric scientists also use this terminology to refer to the region of the lower troposphere immediately above the surface where there is nearly constant potential temperature and specific humidity with height. As in the ocean, this atmospheric zone is characterised by turbulence resulting in a stable vertical temperature profile. Given that the MLD terminology may present confusion for marine scientists, the atmospheric MLD will be referred to as the mixed layer height (MLH) in the revised
15  manuscript, representing the height above the surface of the convective mixed layer or the convective boundary layer.

The MLH is the major part of the marine boundary layer (MBL), which is the height of the atmospheric mixed layer from the ocean surface to a capping inversion, referred to in the manuscript as the entrainment zone. The boundary between the convective mixed layer below and the warmer layer above is marked by the base of the clouds. In Eqn. 1, $H$ describes the
20  mean height of the MBL during each campaign, where as the MLH refers to the height of the MBL at noon for each day during each campaign. Thus, $H$ and MLH are not interchangeable descriptors in the manuscript.

The authors thank the referee for commenting on the manuscript to improve its content.

**Replies to Referee #2**

*Comments on Section 2: Method*
30  *The description of the measurement procedure is insufficient. More details are necessary to understand how atmospheric DMS was measured, e.g. what is the cryogenic trap consist of, is the DMS preconcentrated and trapped before analysis? Is it right that the inlet of the measurement device is around 100 m away from the coral reef? If is it so how you can be sure that the DMSa you measured is directly emitted by the reef.*

35  Reply:
In order to minimise the length of Section 2: 'Methods', a reference is provided to a 2015 publication that gives a complete description of the instrumentation with a detailed analysis of its measurement uncertainty. We appreciate that the referee would like more methodological information provided in this results focussed paper, so it will be included in the revision. Answers to the particular questions posed are as follows: In order to obtain detectable quantities for chromatographic
40  analysis it is necessary to pre-concentrate $DMS_a$ onto a suitable adsorbent, or directly capture it in a cryogenically cooled trap (cryotrap). The cryotrap used with the automated GC-PFPD was constructed by passing 1.6 mm diameter Teflon tubing through ~50 cm of copper tubing of 2.0 mm internal diameter, and bending it into a loop. The cryotrap was immersed in liquid nitrogen during the sample loading period.

45  The inlet for the automated GC-PFPD was positioned at the highest point as close as possible to the coral reef, this being the roof-top of the station laboratory. This inlet was ~100 m from the reef flat on the southern side of the island. DMS was not liberated from the island; the $DMS_a$ measured was derived from the marine environment because DMS is a marine-generated biogenic product. As explained in the manuscript, there was a continuous oceanic $DMS_a$ signal derived from phytoplankton and other pelagic marine biota, while occasional $DMS_a$ spikes were observed that were inconsistent with the usual wind
50  speed driven physical processes that exchange DMS from the ocean surface. This is the objective of the manuscript, i.e. to

explain at length why these spikes could be attributed to DMS emissions from the coral reef. In Section 3 of the manuscript we present a detailed analysis of the accompanying meteorological measurements, tidal information and air parcel back trajectories to provide compelling evidence that these $DMS_a$ spikes came from the coral reef.

5 *Comment:*
*Give more details about how you determined low and high tides. You give even a negative value (p6 line 18). Did you use the height of the reefs as a zero-point?*

Reply:
10 As explained in Section 2.1, tidal information was sourced from underlined{predictions} provided by the National Tidal Unit of the Australian Bureau of Meteorology (BoM). Australian tidal authorities have adopted a 20-year tidal datum epoch from 1992 to 2011 as the basis for calculating tidal planes. When the low water calculation falls below the datum it is given a minus value. Low tide heights given in the manuscript are reported as specified by the BoM. The time of the dry season campaign in 2013 was planned to coincide with the very low (*spring*) low tides that occur in July. It must be understood that tidal
15 heights and times are predictions. The BoM clearly states that tidal predictions for Heron Island are based on limited observations and are, therefore, of secondary quality. The times predicted for high and low tides at this location are thus unlikely to be accurate because of the limited observations. As explained in Section 2.1, several site-specific observations of seawater drainage from the Heron Island reef flat showed that low tides consistently occurred +1.25 h after the predicted times, so tide times were adjusted accordingly. The observed delay from predicted low tide times might be due to the
20 particular geomorphology of the reef flat in combination with possible drainage effects caused by the channel constructed to allow ship access to the island wharf. Accurate specification of the time of low tide was more important than the actual height of the low tide to temporally link our $DMS_a$ measurements at Heron Island.

*Comment:*
25 *In section 2.2 "Flux calculation" you introduced the mass balance equation. Did you perform an error estimation of the different parameter of the equation? Did you estimate the variability of the parameter over time? A mass balance calculation can exhibit many errors due to uncertainties of the different parameter and their variability over time. You have to discuss in more detail that the different parameters you are used are reasonable.*

30 Reply:
Section 2.2 introduces the photochemical ambient mass balance equation (Eq. 1) and the input variables used to calculate $F_{DMS}$. Eq. 1 is applied to estimate the underlined{long-term} underlined{seasonal} DMS emission fluxes during each campaign at Heron Island. The input values entered into Eq. 1 are, therefore, underlined{representative average values}, which dampens out short-term variability. A large part of Section 3.3 in the 'Results and Discussion' is devoted to providing details of how these representative input
35 values were obtained, to show that they are reasonable input values. A propagation of error analysis using the photochemical mass balance approach for DMS air-sea flux has shown that the overall uncertainty in flux estimates is in the range of 31-51% (Avg of 41%, Chen et al., 1999), which is said to compare favourably with other methods. Their sensitivity analysis indicated that $F_{DMS}$ was mainly influenced by the DMS vertical profile and the diel profile for OH. Sensitivity analysis is the investigation of how the uncertainty in the output of a mathematical model or equation can be apportioned to different
40 sources of uncertainty in its inputs. In other words, sensitivity analysis identifies which variables can cause the largest deviations in the outcome. Uncertainty estimations and sensitivity analyses are often run in tandem. In accordance with the uncertainty analysis of Chen et al., (1999) we have quoted an uncertainty of ~50% for the seasonal flux estimates at Heron Island. The following information is provided to satisfy the referee's concerns regarding the variability of each input value in Eq. 1, and will be incorporated into the revised manuscript.
45
1. $[DMS_a]$. It is stated in Section 3.3 that the number of $DMS_a$ measurements is sufficiently large that the mean concentrations for each campaign are expected to be representative of $DMS_a$ in the MBL over Heron Island during the wet and dry seasons. Table 1 shows that the mean and SD for $DMS_a$ during the 2012 wet and 2013 dry seasons is $3.9 \pm 1.5$ ($n = 651$) and $1.3 \pm 1.6$ ($n = 923$) nmol m$^{-3}$, respectively.
50

2. [$DMS_t$]. This is reported to be typically 10% of MBL concentrations. The sensitivity of this variable in Eq. 1 is small. When values for [$DMS_t$] of 5% and 20% of MBL concentrations are entered into Eq. 1, $F_{DMS}$ varies by only 1.4-2.6%.

3. $H$. The average midday mixed layer height (MLH) during the 2012 wet and 2013 dry seasons is 977 m ($\pm$ 231m, range 680 to 1460 m, $n$ = 15 days) and 786 m ($\pm$ 290m, range 346 to 1312 m, $n$ = 19 days), respectively. It is noted that these seasonal values determined using the HYSPLIT model are consistent with measurements made on-site at Heron Island during the June 2009 dry season and February 2010 wet season.

4. [$OH$]. Values of $1.8 \times 10^6$ (2012 wet season) and $1.6 \times 10^6$ molecules cm$^{-3}$ (2013 dry season) were applied according to reported average values over the South Pacific Ocean, in conjunction with a comparison of average solar irradiance we measured at Heron Island in the different seasons.

5. $K$. A value of $6.5 \times 10^{-12}$ cm molecule$^{-1}$ s$^{-1}$ was applied, this being the sum of the abstraction and addition rate reactions of OH with DMS at 25°C and 1 atmosphere pressure, which represents the temperature and pressure during both campaigns. This value for $K$ is a well established value used in atmospheric models.

6. $E_v$. A value of 0.004 m s$^{-1}$ was applied according to average data obtained from Lagrangian experiments in the southern hemisphere remote MBL. This entrainment rate from the lower troposphere into the MBL is typically very low, and when this $E_v$ value is varied by ±100% in Eq. 1 it has a sensitivity effect of 12-19% on $F_{DMS}$.

*Comments on Section 3: Results and Discussion:*
*An overall description of your data is missing. What are the general patterns of your data? Is there a general trend? Additionally, you start directly with the interpretation of the peaks without any introducing sentences. Say in the beginning shorty what you have done and why and what you found.*

Reply:
An introductory paragraph that provides a general description and summary of the results will be included at the beginning of Section 3: 'Results and Discussion' in the revised manuscript. This introductory paragraph will not repeat information given in Section 4: 'Conclusions'.

*Comment:*
*In the first paragraph (p5 l3-10) you mentioned many time points which is hard for the reader to follow. Additionally, the different time points are hard to see in fig. 3. Maybe show clearly in the fig the time steps you described in detail and maybe reword the text a little bit for a better understanding for the reader.*

Reply:
The referee has previously asked how can we be sure that the DMS$_a$ we measured is directly emitted from the reef? The information on P5, L3-10 is provided in sufficient detail to fully describe the circumstances leading to the DMS$_a$ spike shown in Fig.3b, which provides compelling environmental evidence that the spike was derived from the platform reef surrounding Heron Island. This information is not hard to follow if carefully read while also carefully referring to Fig. 3b. When Fig. 3 is viewed at full screen width (e.g. 177% for PDF) the time points described can be clearly seen for reference to the textual description. The authors do not want to add any more detail to Fig. 3, such as notation points or description boxes, which will only serve to clutter the results presented. The information discussed in the manuscript on P5, L3-10 will be reviewed to see if it can be stated any more succinctly without removing any of the details required to fully describe the circumstances leading to the DMS$_a$ spike.

*Comment:*
*P6 L3-15: Why you talked in this paragraph about the measurements on 16 March and before about data from 17 March. Why it is not in chronological order?*

Reply:
The manuscript will be revised so that each $DMS_a$ spike detected during the wet season campaign is discussed in chronological order in separate paragraphs to assist reader interpretation.

*Comment:*
*The mixed layer depth (MLD) you mentioned in the text (p7 line 28) is it in the water or in the atmosphere. Is it the same like the MBL? The MLD is generally used for the water. Please clarify.*

10 Reply:
The referee may be more familiar with the MLD when used in the marine context; however, meteorologists and atmospheric scientists also use this terminology to refer to the region of the lower troposphere immediately above the surface where there is nearly constant potential temperature and specific humidity with height. As in the ocean, this atmospheric zone is characterised by turbulence resulting in a stable vertical temperature profile. Given that the MLD terminology may present
15 confusion for marine scientists, the atmospheric MLD will be referred to as the mixed layer height (MLH) in the revised manuscript, representing the height above the surface of the convective mixed layer or the convective boundary layer. The MLH is the major part of the marine boundary layer (MBL), which is the height of the atmospheric mixed layer from the ocean surface to a capping inversion, referred to in the manuscript as the entrainment zone. The boundary between the convective mixed layer below and the warmer layer above is marked by the base of the clouds.

*Comment:*
*Can you discuss in the results and discussion section the stress level and health conditions of the coral reef you investigated? Is the reef already affected by global change (temperature, pH), has it a high biodiversity, was coral bleaching observed? Can these factors affect the DMSP and DMS production? Are the events observed during the measurements (very low tides,*
25 *reef exposure to the air, rainfall on the corals) normal events which occurred on a regular base or were these extreme and seldom events?*

Reply:
Complementary measurements of the store of DMSP in *Acropora aspera* branching coral during the campaigns in 2012 and
30 2013 indicated that this coral growing on the platform reef surrounding Heron Island was not temperature stressed and appeared to be in good health. This supporting information will soon be reported in Analytical and Bioanalytical Chemistry (doi:10.1007/s00216-016-0141-5). There was no evidence of coral bleaching in the 2012 late summer wet season to affect the usually high biodiversity. A note about this supporting information relating to the health of *Acropora aspera* growing on the Heron Island reef flat will be incorporated into the revised manuscript. The southern GBR has been less affected by
35 warming sea surface temperatures than the northern GBR. This was dramatically shown in the previous 2015-16 summer when a strong *El Niño* Southern Oscillation event enhanced abnormally warm sea surface temperatures, resulting in extensive bleaching to the northern third of the GBR. There was a gradation of coral bleaching mortality, ranging from high in the northern GBR to virtually none on the southern GBR where the Capricorn Bunker Group of coral reefs is situated. We observed a few instances of coral colony bleaching on the Heron Island reef flat in February 2016, which was not observed
40 in March 2012. The GBR Marine Park Authority provides further information about this north to south gradation of coral bleaching during the 2015-16 summer at this web link http://www.gbrmpa.gov.au/media-room/coral-bleaching

Coral reefs are regularly aerially exposed; the extent of that exposure depends on the tidal phase. Very low *spring* tides are experienced in the middle (austral winter) and end of the year around Christmas (austral summer) along the Australian east
45 coast. The very low *spring* tides in July during the 2013 dry season campaign were not unexpected for that time of the year. Rainfall on an aerially exposed coral reef is an unpredictable and irregular event. This is expected to be one of the factors leading to the intermittent nature of the $DMS_a$ spikes detected at Heron Island. As is evident from the entire winter dataset, the intensity of the $DMS_a$ spike detected in the early evening of 25 July 2013 was a unique event, and it was good fortune to be on-site at that time with equipment to detect and quantify it.

50

*Comment:*

*It would be also interesting to measure directly DMS emissions by the corals in incubation experiments under different environmental conditions to have the direct evidence that the DMSa is coming from the corals directly. It is clear that this cannot be part of this study but is interesting to investigate in future studies.*

Reply:

Previously, a number of coral chamber studies have been conducted to investigate the release of DMS from coral into the chamber headspace under varying conditions. The following publications describe laboratory studies where coral (or its endosymbiotic algae) were placed into chambers and DMS emission from the coral was measured:

Fisher and Jones (2012), *Biogeochemistry*, doi:10.1007/s10533-012-9719-y

Deschaseaux et al., (2014), *Journal of Experimental Marine Biology and Ecology*, doi:10.1016/j.jembe.2014.05.018

Deschaseaux et al., (2014), *Limnology and Oceanography*, doi:10.4319/lo.2014.59.3.0758

Swan et al., (2016), *Journal of Atmospheric Chemistry*, doi:10.1007/s10874-016-9327-7

Hopkins et al., (2016), *Scientific Reports*, doi:10.1038/srep36031

In each of these laboratory chamber studies it is apparent that the coral or its algal symbionts were the source of the DMS measured. What is unique about the manuscript we present here for publication in Biogeosciences is that it is the first study conducted on-site at the GBR with sufficient sampling frequency to underline characterize coral reef DMS emissions providing convincing evidence that the coral reef is a source of DMS to the natural atmospheric environment. As stated in the conclusion, what is now required is further on-site continuous sampling of $DMS_a$ at the GBR to more closely examine factors that cause coral reefs to emit DMS to the atmosphere. Chemical ionisation mass spectrometry is recommended because it provides higher temporal resolution than the automated GC we used for the 2012 and 2013 campaigns at Heron Island.

*Comment:*

*Figures: Fig 1 is not necessary to understand the paper and is not discussed in detail in the paper. It has not important new information. I recommend to delete it.*

Reply:

The conceptual model shown in Fig.1a concisely describes the factors and processes controlling $DMS_a$ derived sulfate aerosol production over the GBR. In particular, it shows the oceanic $DMS_a$ source that provided the baseline $DMS_a$ signal shown in Figs 3&6, and how rainfall can induce emissions of DMS from the coral reef at low tide. Fig. 1a also depicts the atmospheric processes leading to formation of CCN, providing scattering of solar radiation back into space. Fig. 1b is referred to on P6, L11 to pictorially describe the Capricorn Bunker Group of coral reefs to the SE of Heron Island, which is important to assist the discussion about the indicated reason for the largest $DMS_a$ spike detected during the 2012 wet season campaign. Another referee of our manuscript has commented that Fig. 1b is a 'great' figure that provides a compelling picture of cloud formation processes in operation over the southern GBR. The authors would like to retain Fig. 1 in the manuscript because we consider that it provides a useful pictorial to support information provided in the introduction and the discussion of results.

*Comment: Figures 3 and 6 are hard to read. The grey, green and blue colors are hard to distinguish and there are too many parameters in one graph. Additionally, you discussed a lot time points but they are hard to see in the sub-panels. See comment above.*

Reply:

As previously explained to the referee, the time points shown in Figs 3&6 can be clearly seen for reference to the textual description when viewed at full screen width (e.g. 177% for PDF). The four colours chosen for $DMS_a$, WS, tide height and rainfall were selected to provide good contrast between each parameter. Figs 3&6 may appear to be "loaded" with data but each of the parameters shown are key to understanding the evolution of the $DMS_a$ spikes and the background $DMS_a$ signal.

5 For example, the alignment of WS with the background $DMS_a$ signal shown in Fig 3a provides a convincing picture that it is the oceanic-derived $DMS_a$ signal because WS is the major factor associated with mass transfer of DMS from the ocean surface to the atmosphere. If any of the four parameters shown in Figs 3&6 were to be removed from the time plots it would be impossible to adequately explain the reasons for the $DMS_a$ spikes detected from the coral reef. This complexity of interacting processes leading to the $DMS_a$ spikes from the coral reef demands that Figs 3&6 highlight these four parameters

10 even if they appear "busy". With this in mind, the extracted time series shown in Figs 3b&c and 6b&c were generated to provide additional clarity of the particular events discussed in the manuscript.

The authors thank the referee for commenting on the manuscript to improve its content.

**List of relevant changes made to revised manuscript**

P11, L10: New reference included

20 P11, L28: New reference included

P12, L18-29: Additional information about of $DMS_a$ sampling methodology included as requested by both referees

P13, L22: The word 'atmospheric' included to emphasise that Eq. 1 relates to the atmospheric environment rather than the

25 marine environment

P13, L31: 'depth' changed to 'height' to respond to comment 8 by Ref#1 and similar comment by Ref#2

P14, L9-26: Introductory paragraph added to respond to comment 5 by Ref#1 and similar comment by Ref#2. New reference

30 included at the end of this paragraph to provide information about the state of health of coral growing on the Heron Island reef flat in the late summer wet season of 2012, as requested by Ref#2

P16, L8-13: Rearrangement of section 3.1 to put all information in chronological order as requested by Ref#2

35 P17, L21-33: Rearrangement of section 3.3 to improve reader interpretation with inclusion of sensitivity analysis information for input variables in Eq. 1

P18, L3-4: Inclusion of sensitivity analysis information for input variable $E_v$ in Eq. 1

40 P18, L6: Clarification of input variable $K$ in Eq. 1

P19, L34: Inclusion of additional funding information

P20-23: References. EndNote X3 was used to insert and format references. The reference listing was moved from the end of

45 the manuscript to P20, so the entire reference listing is marked-up in red in the revised manuscript.

P24, Table 1: ppt mixing ratios have been included with $DMS_a$ molar concentrations to partly respond to comment 4 by Ref#1

[revised manuscript text omitted]